# Replacing the Ex-Def Baseline in AutoML by Naive AutoML

**Felix Mohr**                                          FELIX.MOHR@UNISABANA.EDU.CO
*Universidad de La Sabana, Colombia*

**Marcel Wever**                                    MARCEL.WEVER@UNI-PADERBORN.DE
*Paderborn University, Germany*

## Abstract

Automated Machine Learning (AutoML) is the problem of automatically finding the pipeline with the best generalization performance on some given dataset. AutoML has received enormous attention in the last decade and has been addressed with sophisticated black-box optimization techniques like Bayesian Optimization, Genetic Algorithms, or Tree Search. These approaches are almost never compared to simple baselines to see how much they improve over simple but easy to implement approaches. We present Naive AutoML, a very simple baseline for AutoML that exploits meta-knowledge about machine learning problems and makes simplifying, yet, effective assumptions to quickly come to high-quality solutions. In 1h experiments, state of the art approaches can hardly improve over Naive AutoML, which in turn comes along with advantages such as interpretability and flexibility.

## 1. Introduction

AutoML is the problem of automatically finding the data transformation and learning algorithms with the best generalization performance on a given dataset. The combination of such algorithms is typically called machine learning *pipeline*, because several algorithms for data manipulation and analysis are concatenated sequentially. To optimize such a machine learning pipeline, important decisions do not only include the learning algorithm itself but also its parameters as well as a definition of which features should be used by the learner.

The baselines used to assess the performance of AutoML tools are often other AutoML tools or random search. A simple but perhaps more sensible baseline than random search would be to imitate the steps a human data scientist would take. Without such baselines, we do not learn how AutoML tools improve upon ad-hoc techniques but only how they compare relatively to each other. To our knowledge, the only work accounting for such baselines is (Thornton et al., 2013), using the Exhaustive-Default ("Ex-def") baseline, which is to take the default parametrized model that is best in a cross-validation. They also discuss a grid search, which is however not applicable in practice.

In this paper, we make up for such a baseline, which imitates the process a human expert might go through to find the best pipeline. Instead of crawling an enormous search space and to pick the best one seen within some timeout, the idea is to conduct an analytical process and *derive* the appropriate pipeline from it. In this, we implicitly assume a form of independence of the optimization decisions and hence act a bit *naive* w.r.t. potential interactions between them; this is why we call the approach Naive AutoML. In this sense, we complement the "Ex-def" baseline (Thornton et al., 2013) by what a somewhat more experienced data scientist would maybe do.

The surprising result of our experimental evaluation is that, for runtimes of 1h, the black box optimizers are hardly ever able to improve upon Naive AutoML. In fact, the experiments even show that the "Ex-def" baseline itself is already quite strong in this time frame. This is not a contradiction to the results in (Thornton et al., 2013), where a timeout of 30 hours was used. While this observations calls for more exhaustive experiments, it is already evident that simple baselines are stronger than perhaps believed.

Even from these results, we can learn two things. First, it seems sensible to adopt Naive AutoML as a standard baseline for the evaluation of AutoML tools in order to demonstrate their superiority over a very simple yet often effective approach. Second, the success of Naive AutoML challenges the view of black box optimization for AutoML in general and invites to consider more specific approaches taylored for AutoML.

## 2. Problem Definition

In this paper, we are focused on AutoML for supervised learning. Formally, in the supervised learning context, we assume some *instance space* $\mathcal{X} \subseteq \mathbb{R}^d$ and a *label space* $\mathcal{Y}$. A *dataset* $D \subset \{(x, y) \mid x \in \mathcal{X}, y \in \mathcal{Y}\}$ is a *finite* relation between the instance space and the label space, and we denote as $\mathcal{D}$ the set of all possible datasets. We consider two types of operations over instance and label spaces:

1. *Transformers.* A transformer is a function $t : \mathcal{X}_A \to \mathcal{X}_B$, converting an instance $x$ of instance space $\mathcal{X}_A$ into an instance of another instance space $\mathcal{X}_B$.

2. *Predictors.* A predictor is a function $p : \mathcal{X}_p \to \mathcal{Y}$, assigning an instance of its instance space $\mathcal{X}_p$ a label of the original label space $\mathcal{Y}$.

In this paper, a *pipeline* $P = t_1 \circ .. \circ t_k \circ p$ is a functional concatenation in which $t_i : \mathcal{X}_{i-1} \to \mathcal{X}_i$ are transformers with $\mathcal{X}_0 = \mathcal{X}$ being the original instance space, and $p : \mathcal{X}_k \to \mathcal{Y}$ is a predictor. Hence, a pipeline is a function $P : \mathcal{X} \to \mathcal{Y}$ that assigns a label to each object of the instance space. We denote as $\mathcal{P}$ the space of all pipelines of this kind. In general, the first part of a pipeline could not only be a sequence but also a *transformation tree* with several parallel transformations that are then merged (Olson and Moore, 2016), but we do not consider such structures in this paper since they are not necessary for our key argument.

In addition to the sequential structure, many AutoML approaches restrict the search space still a bit further. This is by putting an order on the pre-processing steps and allowing only on transformer per type. So $\mathcal{P}$ will only contain pipelines compatible with this order. We can then express *every* element of $\mathcal{P}$ as a concatenation of $k + 1$ functions, where $k$ is the number of considered transformation algorithm types.

In this space of possible pipelines, the goal is to find the one that optimizes a pre-defined performance measure $\phi : \mathcal{D} \times \mathcal{P} \to \mathbb{R}$. Typical measures are the error rate, log-loss or, for the case of binary classification, AUC/ROC or the F1-measure on a given dataset.

## 3. Naive AutoML

### 3.1 Naivety Assumptions

Naive AutoML makes, among others, the assumption that the optimal pipeline is the one that is locally best for each of its transformers and the final predictor. In other words,

taking into account pipelines with (up to) $k$ transformers and a predictor, we assume that for all datasets $D$ and all $1 \leq i \leq k+1$

$$c_i^* \in \arg\min_{c_i} \phi(D, c_1 \circ .. \circ c_{k+1})$$

is *invariant* to the choices of $c_1, ..c_{i-1}, c_{i+1}, .., c_{k+1}$, which are supposed to be fixed in the above equation. Note that we here use the letter $c$ instead of $t$ for transformers or $p$ for the predictor, because $c$ may be any of the two types.

We dub the approach Naive AutoML, because there is a direct link to the assumption made by the Naive Bayes classifier. Consider $\mathcal{P}$ an urn and denote as $Y$ the event to observe an optimal pipeline in the urn. Then

$$\mathbb{P}(Y \mid c_1, .., c_{k+1}) \propto \mathbb{P}(c_1, .., c_{k+1} \mid Y)\mathbb{P}(Y) \stackrel{naive}{=} \mathbb{P}(c_i \mid Y) \prod_{j=1, j \neq i}^{k+1} \mathbb{P}(c_j \mid Y)\mathbb{P}(Y),$$

in which we consider $c_j$ to be fixed components for $j \neq i$, and only $c_i$ being subject to optimization. Applying Bayes theorem again to $\mathbb{P}(c_i \mid Y)$ and observing that the remaining product is a constant regardless the choices of $c_{i \neq j}$, it gets clear that the optimal solution is the one that maximizes the probability of being locally optimal, and that this choice is *independent* of the choice of the other components.

The typical approach to optimize the $c_i$ is not to directly construct those functions but to adopt parametrized model building processes that create these functions. For example, $c_1$ could be a projection obtained by determining some features which we want to stay with, or $c_{k+1}$ could be a trained neural network. These induction processes for the components can be described by an algorithm $a_i$ and a hyper-parametrization $\theta_i$ of the algorithm. The component $c_i$ is obtained by running $a_i$ under parameters $\theta_i$ with some training data. So to optimize $c_i$, we need to optimally choose $a_i$ and $\theta_i$.

Within this regime, Naive AutoML makes the additional (maybe not always correct) assumption that even each component $c_i$ can be optimized by local optimization techniques. More precisely, it is assumed that the algorithm that yields the best component when using the default parametrization is also the algorithm that yields the best component if all algorithms are run with the best parametrization possible.

### 3.2 A Prototype for Naive AutoML

In this paper, we work with a specific *exemplary* realization scheme of Naive AutoML. However, this scheme is *not* what we refer to as Naive AutoML but only *one prototypical instantiation*. We certainly do not claim this prototype to be the last answer in Naive AutoML; better schemes can be found in future work.

Our prototype scheme consists of six *stages* and is sketched in Fig. 1. The first stage simply cross-validates every learning algorithm once without setting any of its hyperparameters. This stage corresponds to the "Ex-def" baseline used in (Thornton et al., 2013). The second stage simply pairs all feature scalers with all learners and observes their performance. The third stage is independent of the first two stages and adopts filtering techniques to identify a subset of the features that are expected to bring the best performance (on average). The fourth stage combines all previous candidates with each homogeneous

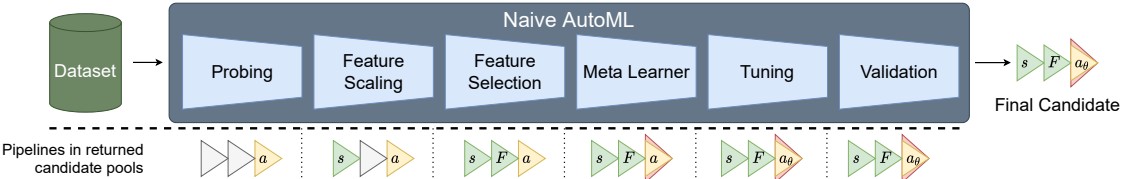

Figure 1: Prototype of a six-stage Naive AutoML process (top), taking a dataset as an input and returning a final candidate. The lower part of the figure illustrates the potential shapes of machine learning pipelines contained in the candidate pool after the execution of the respective stage.

meta learner, and the fifth stage tunes the *learner* parameters of the incoming candidates (with a given time budget). In the final stage, the most promising candidates are evaluated against a validation dataset. The stages are described in more detail in the Appendix.

This stage scheme produces pipelines with up to three components. A candidate is encoded by a tuple $(s, F, a, \theta)$, where $s$ encodes the feature transformations (applied to all columns), $F$ is a *set* of feature indices to consider, $a$ is the prediction algorithm, and $\theta$ the parametrization for $a$. The reason why $s$ is a function but $F$ is a set is simply that the feature selection is a projection that is entirely described by the features it shall retrain, but the scaling implies a functional transformation of data that cannot be captured so easily.

Despite being only an example scheme, one advantage of Naive AutoML that already becomes clear here is that it directly generates important insights that can significantly support the data scientist working with it. For example, a question like "what is the potential of feature selection on the given data?" can be answered by black-box approaches only after some post-processing, if at all. In our scheme, the filtering stage (cf. Sec. A.3) is a very good basis to give an initial answer to this question. In this sense, Naive AutoML presents itself as more amenable to the growing demand for meaningful *interaction* between the tool and the human (Wang et al., 2019; Crisan and Fiore-Gartland, 2021; Drozdal et al., 2020; Wang et al., 2021) compared to the currently adopted black-box approaches.

## 4. Evaluation

In this section, we want to address the following question: *How much can state-of-the-art tools for Python and Java improve over the baseline imposed by Naive AutoML?*

We stress our point of view that Naive AutoML is in fact the baseline here and not the competing technique. Naive AutoML *updates* the previous "Ex-def" baseline by proposing a more sophisticated data scientist (acknowledging that an expert data scientist would be even more flexible and hence might even be stronger than our Naive AutoML).

### 4.1 Compared Algorithms

We ran Naive AutoML in two configurations. We refer to "primitive" as the version of Naive AutoML that *only* adopts the probing (first) stage and nothing else. Once again,

this corresponds to the "Ex-def" baseline proposed in (Thornton et al., 2013). The profile containing *all* the stages (including validation), is referred to as "full".

On the state-of-the-art side, we compare solutions with competitive performance for both WEKA (Hall et al., 2009) and scikit-learn (Pedregosa et al., 2011). To our knowledge, ML-Plan is the best performing AutoML tool for WEKA to date, but we also consider Auto-WEKA to contrast its performance against the "Ex-def" baseline. On the scikit-learn side, we consider auto-sklearn as a competitor (without warm-starting and without final ensemble building). We consider version 0.12.0, which underwent substantial changes and improvements compared to the original version proposed by (Feurer et al., 2015).

### 4.2 Experiment Setup

The evaluation is based on the dataset portfolio proposed in (Gijsbers et al., 2019). This is a collection of datasets available on openml.org (Vanschoren et al., 2013). To complement these datasets, we have added some of the datasets that were used in the Auto-WEKA paper (Thornton et al., 2013) and have been used frequently for comparison in publications on AutoML. Five datasets of (Gijsbers et al., 2019) (23, 31, 188, 40996, 41161, 42734) were removed due to technical issues in the data loading process.

For each dataset, 10 random train-test splits were created, and all algorithms were run once on each of these splits, using the train data for optimizing, and the test data to assessing the performance. Needless to say, the splits were the same for all approaches.

For all algorithms, we allowed a total runtime of 1h, and the runtime for a single pipeline execution was configured to take up to 1 minute. In Auto-WEKA, it is not possible to configure the maximum runtime for single executions; this variable is interenally controlled. In Naive AutoML, we imposed stage time bounds for the meta and the parameter tuning stages of 5 minutes respectively (the other stages were not equipped with a dedicated timeout). Of course, on hitting the time bound of 1h, Naive AutoML was stopped regardless the phase in which it was, and the best seen solution was returned.

Computations were run on Linux machines, each of them equipped with 2.6Ghz Intel Xeon E5-2670 processors and 32GB memory. In spite of the technical possibilities, we did *not* parallelize evaluations, i.e. all the tools were configured to run with a single CPU core.

### 4.3 Results

Due to space limitations, Fig. 2 only summarizes the results very superficially through final ranks. More detailed results can be found in Table 2 and Fig. 4 in Sec. C of the appendix. The code (both Java and Python) to reproduce these results is publicly available[1].

The results show that the cases in which any of the advanced AutoML tools achieves a statistically significant improvement form a small minority. In fact, there are even quite some datasets on which Naive AutoML performs *better*. Auto-WEKA is *never* better than the "primitive" Naive AutoML approach ("Ex-def") and even worse in 6 cases. The performance of ML-Plan is only marginally better in the comparison with Naive AutoML but also does not show a substantial improvement over it and is outperformed even 11 times by "Ex-def". The comparison between auto-sklearn and the "full" Naive AutoML profile, we

---

1. `https://github.com/fmohr/naiveautoml/tree/icml2021`

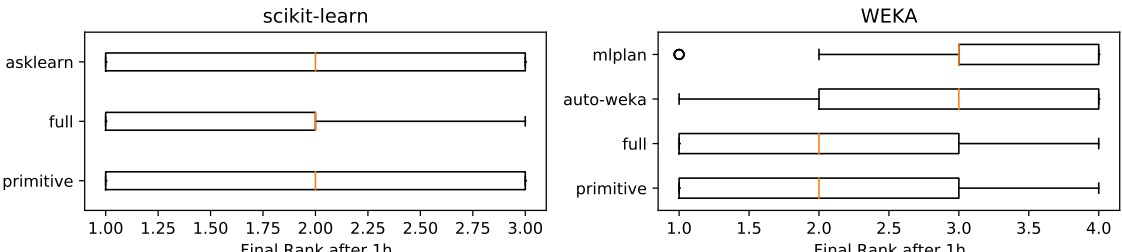

Figure 2: Final performance ranks after 1h runtime. See Appendix for details.

can see that it substantially improves upon Naive AutoML in only five out of 67 cases and is outperformed 8 times. Among the five cases of improvement, auto-sklearn uses in four of the cases algorithms not supported by Naive AutoML in the current version.

The observations on the "primitive" profile can be put into words by saying that if we run a simple `for` loop over the possible base learners (using default parameters and ignoring meta-learners or any kind of feature transformation), then we obtain an equal or better performance than Auto-WEKA, ML-Plan, or auto-sklearn in more than 90% of the cases; no pre-processors are required. The results for the "primitive" solution arrive often in the range of some few minutes. So they are cheap to obtain, and other approaches rarely can improve upon it even when running for an hour.

In our view, there are two possible explanations for the missing superiority of the black-box optimization techniques. The first is simply that the global optimum *is* in fact obtained by choosing the best algorithm without any further optimizations. That is, there is simply no *potential* for optimization. On some datasets (with almost perfect performance) this is clearly the case. The second is that there *maybe* is potential for optimization, but the resource limitations impede that the black-box approaches can develop their full potential. Theoretically, all of the black-box approaches converge to the globally optimal solution. However, only a tiny fraction of the search space can be examined in any reasonable time-out, and the number of evaluations that can be made in that time are not enough to learn enough about the performance landscape to steer the search process in a meaningful way. It is hence worthwhile to run the experiments with higher timeouts to see whether the AutoML tools can eventually outperform Naive AutoML and after which time.

## 5. Conclusion

This paper proposes Naive AutoML, a slightly more sophisticated baseline for AutoML than proposed with "Ex-def" in (Thornton et al., 2013). Instead of searching a complex pipeline search space with a black-box optimizer, Naive AutoML imitates the sequential workflow conducted by a data scientist, which *implicitly* defines a solution pipeline. We empirically demonstrate that state-of-the-art tools are rarely able to outperform this baseline (sometimes the contrary is true). Producing competitive results to the state of the art, Naive AutoML is not only more transparent and understandable to the expert but also more flexible, because all modifications can be directly realized in the stage implementations instead of having it to be injected into the solver through the problem encoding.

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

# Appendix

## Appendix A. Details on the Stages of the Naive AutoML Prototype

A candidate is encoded by a tuple $(s, F, a, \theta)$, where $s$ encodes the feature transformations (applied to all columns), $F$ is a *set* of feature indices to consider, $a$ is the prediction algorithm, and $\theta$ the parametrization for $a$. The symbol $\bot$ is used to denote that a part of the description remains empty.

In the following sub-sections, we assume that candidates are evaluated with some standard evaluation scheme. A typical choice to evaluate candidates is to use standard cross validation techniques. For example, in the evaluation in Sec. 4, we conduct a Monte-Carlo Cross Validation (MCCV) with a train fold size of 70% and 5 repetitions. This means that we build 5 random splits of 70% training data and 30% validation data each, train and test over the 5 folds using the desired metric, and then average these observations to get to a score. However, Naive AutoML is not committed to a particular type of scoring function and could, for example, also be run with a 10-fold cross validation. We hence just shall assume that every pipeline has some *score*, and we leave it to the concrete implementation to implement one or another method.

### A.1 The Probing Stage

This stage corresponds to the "Ex-def" suggested in the evaluation of (Thornton et al., 2013). Formally, we create and evaluate *all* candidates of the form $(\bot, \bot, a, \bot)$, where $a$ is one of the available *base* prediction algorithms. Here, we do not consider ensemble learners like Boosting or Bagging but only those that are already implemented with a specific base learner, such as Random Forests.

### A.2 The Feature-Scaling Stage

In this step, we examine the benefit of different feature scaling operations for the base learners. To this end, two or three cheap distance-sensitive pilot classifiers like kNN or SVMs are used to assess the impact of different feature scaling techniques such as (mean)-normalization, standardization (mean 0 and std 1), or quantile-based re-scaling. In addition, it can make sense to add as a pilot the one or two best candidates resulting from the probing stage. Formally, for each scaler $s$ and each pilot algorithm $a_p$, we evaluate the candidate $(s, \bot, a_p, \bot)$.

For each scaling technique $s$, if at least one pilot classifier improves upon its result of the original data, we evaluate *all* other candidates $(s, \bot, a, \bot)$ as well, where $a$ is any non-pilot base learner.

### A.3 The Filtering Stage (Classifier-Independent Attribute Selection).

This stage consists of two steps. In the first step, we compute the set $F$ of features that are considered relevant. In the second step, we examine how the previously created candidates behave when using only the features of $F$. That is, for every candidate $(s, \bot, a, \bot)$, we create and evaluate the candidate $(s, F, a, \bot)$. Here $s$ may be $\bot$ as well, and the candidates are created in the order of the performance of the pipeline without feature selection. Typ-

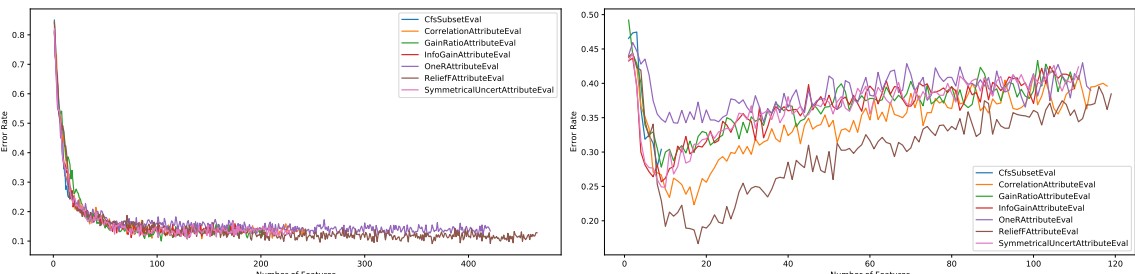

Figure 3: Error rates when applying filtering on the datasets cnae-9 (left) with 856 features in total and Madelon (right) with 500 features in total. Each curve shows at point $x$ the error rate obtained when using the first $x$ attributes according to the ranking created by the respective feature evaluator. In both cases, it is evident that there is little point in using all the features. In the better case, this only implies a waste of computational time (cnae-9) while in the worse case this even implies inferior results (Madelon).

ically, one may want to define a timeout for this stage to not evaluate overly unpromising candidates if a lot of time has already been used.

The feature set $F$ is computed based on the results of different *filtering* techniques (Hall, 1999). Different techniques to *rank* the relevance of features exist. In contrast to *wrapping* (Kohavi and John, 1997), filtering techniques do not adopt (costly) learners to judge feature relevance and hence are typically cheap to compute. We execute all such available filters, which gives us a set of rankings. Then, for each of these rankings, we compute the performance when using the first $l$ features where $l$ is increased until the performance starts to decrease. To assess the performance, a pilot classifier is evaluated on the respective feature set in some kind of cheap cross-validation. In this sense, Naive AutoML uses existing pre-processors only as a source of *suggestions* for candidate feature sets.

To motivate this procedure, it is worth to have a look at the *performance curves* one obtains for the different filters. For each number $x$ of features, we can plot the performance of some pilot learner if using only the first $x$ features in the ranking created by some filter. Fig. 3 illustrates these curves for two datasets and using different filters from WEKA. In the case of the cnae-9 dataset (left), we can see that there is only marginal improvement after $x = 100$, and for the madelon dataset (right) we can even observe that performance *decreases* quickly. In both cases, it is evident that only a relatively small portion of the features is needed to obtain the same or even better results compared to when all features are used (notice that the plot curves to not even show the full range of features for both datasets).

Note that the filtering stage optimizes the feature set $F$ independently of the outcomes of earlier stages. Neither the results of the probing stage nor those of the scaling stage are used in the determination of $F$. Those outcomes are only relevant in that they define the candidates to which the feature selection $F$ should be added, and their order for the evaluation.

One important additional use case for the analysis of such curves is the reduction of evaluation times during the AutoML process. Even if we do not *need* to reduce the dimensionality to obtain good results as in the case of madelon (right), we can often still *approximate* the prediction performance on the full feature set sufficiently well. For example, in the case of cnae-9 (left), we only need to use 100 of the 856 features to obtain comparable results, which should be good enough to steer the search process. Since we know that the runtime of many learners often increases super-linearly (sometimes quadratically) in the number of attributes (Mohr et al., 2021), there is a huge potential in runtime reduction. For example, the time to train a random forest on cnae-9 on some reference machine is 2.2s on average but only 0.5s on 100 attributes with almost the same outcome.

### A.4 Meta-Learner Stage.

In this stage, each candidate of the input pool is taken and used as a template to derive new pipelines in which the base learner is wrapped into a meta-learner. The feature transformation algorithms, if present, are not touched in this stage.

The algorithms we consider here as meta-learners are also sometimes called (*homogeneous*) ensembles. The idea of those learners is to take several copies of a base learner and somehow combine them into a new augmented learner. Typical examples are Bagging (Breiman, 1996) and AdaBoost (Freund and Schapire, 1998). This is opposed to *heterogeneous* ensembles like Stacking (Wolpert, 1992) or majority vote ensembles, in which different learners of different types are combined. Since we augment existing pipelines (with one learner), we only work with homogeneous ensembles here.

Forming a heterogeneous ensemble is a reasonable final step. This would be identical to the strategy pursued in (Feurer et al., 2015) to eventually take the best $k$ learners seen so far and merge them into a voting ensemble. Alternatively, more sophisticated approaches could try to optimize such an ensemble using previous observations.

### A.5 Parameter Tuning.

This stage simply tries to find better hyperparameters for the predictor in one or more candidate pipelines. Given a total timeout for this phase, the candidates are optimized in the order of their performance with a (local) timeout and a maximum number of evaluations. For algorithms with a small parameter space, all candidates can be enumerated, e.g. k-nearest neighbors with some reasonable candidate set for the number of neighbors. For all other algorithms, standard techniques for finding good configurations such as SMAC (Hutter et al., 2011), Hyperband (Li et al., 2017), etc. can be employed. In this paper, we even only adopt a simple random search.

### A.6 (Validation-Based) Model Selection.

The default decision of Naive AutoML to select the best seen pipeline might not always be the best. Despite the simplicity and naivity of Naive AutoML, there is some significant optimization going on, potentially leading to over-fitting.

The potential need of some validation has been recognized earlier (Mohr et al., 2018), and we adopt a similar strategy here. More precisely, the idea is to keep back a certain

portion of the original data that is not shown to the optimization process and *only* used in this final stage. The main difference to (Mohr et al., 2018) is that we do not add the validation data to the pool and then run a cross-validation on the augmented dataset, but we here simply conduct a "classical" single-fold validation on this hold-out set.

The data portion used for validation can be chosen dynamically based on the desired guarantees on the generalization performance. For a set of $m$ remaining competitive candidates, the Hoeffding bound allows us to estimate the out-of-sample error with $\mathbb{P}(|\mu - \nu| > \varepsilon) < 2me^{-2\varepsilon n}$, where $\mu$ is the true out-of-sample error, $\nu$ is the error on the validation fold, and $n$ is the *size* of the validation fold. Given sufficient data and for a moderate number $m$, say $m = 10$, this can be quite a good bound and impose an important remedy against over-fitting after an exhaustive optimization effort.

Unfortunately, the number of validation samples available is often not sufficient to make the Hoeffding bound meaningful. In most cases, we want to choose $\varepsilon \approx 0.01$ and have the bound relatively small, say, 0.1. To assure such a bound for even only one candidate (effectively *testing* its performance), the validation fold must already contain roughly 15000 examples. For $m = 10$, we would need 35,000 validation instances, which are often not available.

We hence propose to use both the "internal" score observed during the optimization process *and* the validation score for model selection and weight the two scores based on the validation set size. To this end, we introduce a parameter $\bar{n}$ that quantifies the number of validation instances required to exclusively use validation instances to compute the score. Intuitively, $\bar{n}$ is the number of instances needed to get the desired certainty in the Hoeffding bound, e.g., 35,000 instances. The *satisfaction* of the Hoeffding bound can then be expressed as

$$\tau(n) = \min\{1, \frac{n}{\bar{n}}\},$$

where $n$ is the actual size of the validation fold. However, a close-to-0 satisfaction does not necessarily mean a close-to-0 weight of the validation score. For example, if we have 500 instances in total and use 100 of them for validation, then 100 is probably far away from $\bar{n}$, and the satisfaction is almost 0. Still, the 100 instances are a valuable complement to the 400 instances used for training. In this case, it makes more sense to weight the validation score based on the *ratio* between the number of validation instances and totally available instances; here, this would be 0.2. Hence, instead of using the satisfaction of the validation fold size directly as a weight, we use it to determine the point on a linear scale between the above ratio and 1. Formally, we define the pipeline score in the validation phase then to be

$$\phi_{final}(P) = \phi_{int}(P)(1 - \omega(n)) + \phi_{validate}(P)\omega(n),$$

where $\phi_{int}$ is the internal score obtained using only the data available for optimization, $\phi_{validate}$ is the performance obtained using *only* the validation data (the pipeline has still been defined not using these data points), $n$ is the number of instances in the validation fold, and

$$\omega(n) = \tau(n) + \frac{n}{N}(1 - \tau(n)).$$

In the last term, $N$ is the total number of instances available.

## Appendix B. Datasets

All datasets are available via the openml.org platform (Vanschoren et al., 2013).

## Appendix C. Result Tables for Naive AutoML

The results of the experiments are shown in Table 2. The reported metric is the *error rate*. For each dataset and each approach, we report the *trimmed* mean (10% trimmed) with standard deviation. Nan entries are caused by memory overflows. We refrain from the now somewhat common practice of reporting average ranks, because, in our view, those obscure a lot of important details in the comparison, e.g. on *which* (and *how many*) datasets which algorithm is better than another and by which *margin*.

The formatting semantics of the table is as follows. The best entries (w.r.t to the trimmed mean) per machine learning library are formatted in bold, and those whose result distribution cannot be said to be statistically different to the best one (according to a Wilcoxcon signed rank test with confidence 0.05) or where the performance difference is "irrelevant" ($< 0.01$) are underlined. With respect to the latter one we of course understand that in some cases such marginal difference *can* be relevant, but here we still treat them as equally good to give more significance to the symbols used to denote improvements. For each of the state-of-the-art techniques, we use the ● or ○ symbol to indicate a score that is "substantially" better or worse than the one obtained with Naive AutoML. The symbols are used twice, once to compare against the "primitive" profile and once comparing against the "full" profile. As above, substantial here means that it is not only statistically significant but also that the absolute improvement is at least 0.01. The symbol † is used instead of a ● in cases in which auto-sklearn constructed pipelines with feature transformation algorithms not supported in the algorithm scheme of Naive AutoML; examples are the PCA or feature encodings based on trees. Based on these results, we can now answer the three research questions above.

| openmlid | name | instances | features | classes |
|---|---|---|---|---|
| 3 | kr-vs-kp | 3196 | 37 | 2 |
| 12 | mfeat-factors | 2000 | 217 | 10 |
| 54 | vehicle | 846 | 19 | 4 |
| 181 | yeast | 1484 | 9 | 10 |
| 1049 | pc4 | 1458 | 38 | 2 |
| 1067 | kc1 | 2109 | 22 | 2 |
| 1111 | KDDCup09_appetency | 50000 | 231 | 2 |
| 1457 | amazon-commerce-reviews | 1500 | 10001 | 50 |
| 1461 | bank-marketing | 45211 | 17 | 2 |
| 1464 | blood-transfusion-service-center | 748 | 5 | 2 |
| 1468 | cnae-9 | 1080 | 857 | 9 |
| 1475 | first-order-theorem-proving | 6118 | 52 | 6 |
| 1485 | madelon | 2600 | 501 | 2 |
| 1486 | nomao | 34465 | 119 | 2 |
| 1487 | ozone-level-8hr | 2534 | 73 | 2 |
| 1489 | phoneme | 5404 | 6 | 2 |
| 1494 | qsar-biodeg | 1055 | 42 | 2 |
| 1515 | micro-mass | 571 | 1301 | 20 |
| 1590 | adult | 48842 | 15 | 2 |
| 4134 | Bioresponse | 3751 | 1777 | 2 |
| 4135 | Amazon_employee_access | 32769 | 10 | 2 |
| 4534 | PhishingWebsites | 11055 | 31 | 2 |
| 4538 | GesturePhaseSegmentationProcessed | 9873 | 33 | 5 |
| 4541 | Diabetes130US | 101766 | 50 | 3 |
| 23512 | higgs | 98050 | 29 | 2 |
| 23517 | numerai28.6 | 96320 | 22 | 2 |
| 40498 | wine-quality-white | 4898 | 12 | 7 |
| 40668 | connect-4 | 67557 | 43 | 3 |
| 40670 | dna | 3186 | 181 | 3 |
| 40685 | shuttle | 58000 | 10 | 7 |
| 40701 | churn | 5000 | 21 | 2 |
| 40900 | Satellite | 5100 | 37 | 2 |
| 40975 | car | 1728 | 7 | 4 |
| 40978 | Internet-Advertisements | 3279 | 1559 | 2 |
| 40981 | Australian | 690 | 15 | 2 |
| 40982 | steel-plates-fault | 1941 | 28 | 7 |
| 40983 | wilt | 4839 | 6 | 2 |
| 40984 | segment | 2310 | 20 | 7 |
| 41027 | jungle_chess_2pcs_raw_endgame_complete | 44819 | 7 | 3 |
| 41138 | APSFailure | 76000 | 171 | 2 |
| 41142 | christine | 5418 | 1637 | 2 |
| 41143 | jasmine | 2984 | 145 | 2 |
| 41144 | madeline | 3140 | 260 | 2 |
| 41145 | philippine | 5832 | 309 | 2 |
| 41146 | sylvine | 5124 | 21 | 2 |
| 41147 | albert | 425240 | 79 | 2 |
| 41150 | MiniBooNE | 130064 | 51 | 2 |
| 41156 | ada | 4147 | 49 | 2 |
| 41157 | arcene | 100 | 10001 | 2 |
| 41158 | gina | 3153 | 971 | 2 |
| 41159 | guillermo | 20000 | 4297 | 2 |
| 41162 | kick | 72983 | 33 | 2 |
| 41163 | dilbert | 10000 | 2001 | 5 |
| 41164 | fabert | 8237 | 801 | 7 |
| 41165 | robert | 10000 | 7201 | 10 |
| 41166 | volkert | 58310 | 181 | 10 |
| 41167 | dionis | 416188 | 61 | 355 |
| 41168 | jannis | 83733 | 55 | 4 |
| 41169 | helena | 65196 | 28 | 100 |
| 42732 | sf-police-incidents | 2215023 | 10 | 2 |
| 42733 | Click_prediction_small | 39948 | 12 | 2 |

Table 1: Overview of datasets used in the evaluation.

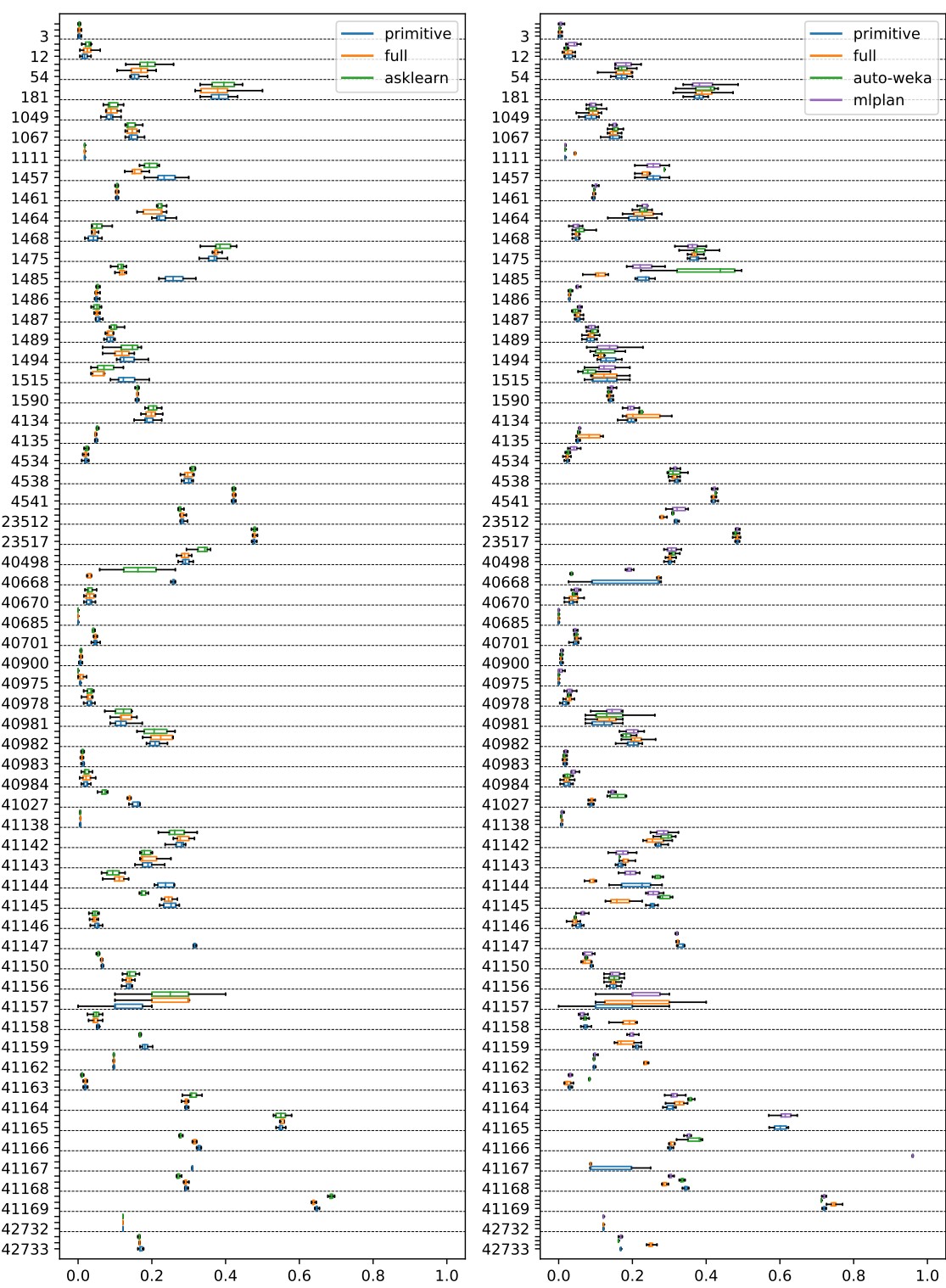

Figure 4: Error rates on the different datasets. Left for scikit-learn and right for WEKA.

Table 2: Mean error rates and standard deviations per machine learning library.

| id | WEKA backend | | | | scikit-learn backend | | |
|---|---|---|---|---|---|---|---|
| | auto-weka | mlplan | primitive | full | asklearn | primitive | full |
| 3 | **0.0±0.0** | 0.01±0.01 | **0.0±0.0** | **0.0±0.0** | **0.0±0.0** | **0.0±0.0** | **0.0±0.0** |
| 12 | **0.02±0.0** | 0.04±0.01 | 0.03±0.01 | 0.03±0.01 | 0.03±0.02 | **0.02±0.02** | 0.03±0.02 |
| 54 | 0.18±0.02 | 0.18±0.03 | **0.17±0.02** | 0.18±0.05 | 0.19±0.03 ○ ○ | **0.16±0.03** | 0.17±0.03 |
| 181 | 0.39±0.04 | 0.39±0.04 | **0.38±0.03** | 0.39±0.04 | 0.39±0.04 | 0.38±0.03 | **0.37±0.05** |
| 1049 | 0.1±0.02 | **0.09±0.01** | 0.09±0.02 | **0.09±0.02** | **0.09±0.02** | 0.09±0.02 | **0.09±0.01** |
| 1067 | **0.15±0.01** | **0.15±0.01** | 0.15±0.02 | **0.15±0.01** | **0.14±0.02** | 0.15±0.02 | 0.15±0.01 |
| 1111 | **0.02±0.0** | **0.02±0.0** ● | **0.02±0.0** | 0.04±0.0 | **0.02±0.0** | **0.02±0.0** | **0.02±0.0** |
| 1457 | 0.29±0.0 | 0.25±0.03 | 0.25±0.03 | **0.24±0.03** | 0.2±0.02 ● ○ | 0.23±0.04 | **0.16±0.03** |
| 1461 | **0.1±0.0** | **0.1±0.0** | **0.1±0.0** | **0.1±0.0** | **0.11±0.0** | **0.11±0.0** | **0.11±0.0** |
| 1464 | 0.22±0.03 | 0.24±0.01 ○ | **0.21±0.04** | 0.23±0.03 | 0.22±0.02 | 0.22±0.02 | **0.21±0.03** |
| 1468 | 0.06±0.02 | **0.05±0.02** | 0.05±0.01 | 0.05±0.01 | 0.05±0.02 | 0.04±0.02 | 0.04±0.01 |
| 1475 | 0.38±0.03 | **0.36±0.02** | 0.37±0.02 | 0.37±0.02 | 0.39±0.03 ○ ○ | **0.37±0.02** | **0.37±0.02** |
| 1485 | 0.39±0.1 ○ ○ | 0.23±0.05   ○ | 0.23±0.02 | **0.11±0.02** | **0.11±0.02** † | 0.26±0.03 | 0.12±0.01 |
| 1486 | **0.03±0.0** | 0.05±0.01 ○ ○ | **0.03±0.0** | 0.03±0.0 | **0.05±0.0** | **0.05±0.0** | **0.05±0.0** |
| 1487 | **0.05±0.01** | 0.06±0.01 | **0.05±0.01** | **0.05±0.01** | **0.05±0.01** | **0.05±0.01** | **0.05±0.01** |
| 1489 | **0.09±0.01** | **0.09±0.01** | **0.09±0.01** | **0.09±0.01** | 0.1±0.01 | **0.09±0.01** | **0.09±0.01** |
| 1494 | 0.13±0.03 | 0.13±0.04 | 0.13±0.02 | **0.12±0.03** | 0.14±0.03   ○ | 0.13±0.03 | **0.12±0.03** |
| 1515 | **0.09±0.03** | 0.13±0.04 | 0.13±0.04 | 0.13±0.04 | **0.07±0.03** ● | 0.13±0.03 | **0.07±0.26** |
| 1590 | **0.14±0.0** | **0.14±0.01** | **0.14±0.0** | **0.14±0.01** | **0.16±0.0** | **0.16±0.0** | **0.16±0.0** |
| 4134 | 0.23±0.02 | **0.2±0.01** | **0.2±0.01** | 0.22±0.05 | 0.2±0.01 | **0.19±0.02** | 0.2±0.02 |
| 4135 | **0.05±0.0** | 0.06±0.01 | **0.05±0.0** | 0.08±0.03 | **0.05±0.0** | **0.05±0.0** | **0.05±0.0** |
| 4534 | 0.03±0.0 | 0.04±0.01 ○ ○ | **0.02±0.0** | 0.02±0.01 | **0.02±0.0** | 0.02±0.01 | **0.02±0.0** |
| 4538 | 0.32±0.02 | 0.32±0.01 | 0.32±0.01 | **0.31±0.01** | 0.31±0.01 | **0.3±0.01** | **0.3±0.01** |
| 4541 | 0.43±0.0 | **0.42±0.01** | **0.42±0.01** | **0.42±0.01** | **0.42±0.0** | **0.42±0.0** | **0.42±0.0** |
| 23512 | 0.31±0.0   ○ | 0.32±0.02   ○ | 0.31±0.02 | **0.28±0.01** | **0.28±0.0** | **0.28±0.01** | **0.28±0.01** |
| 23517 | **0.48±0.01** | **0.48±0.01** | **0.48±0.0** | **0.48±0.01** | **0.48±0.01** | **0.48±0.0** | **0.48±0.0** |
| 40498 | 0.31±0.01 | 0.31±0.02 | **0.3±0.01** | **0.3±0.01** | 0.34±0.02 ○ ○ | **0.29±0.01** | **0.29±0.01** |
| 40668 | **0.03±0.0** ● | 0.19±0.01   ● | 0.21±0.11 | 0.27±0.0 | 0.17±0.06 † ○ | 0.26±0.0 | **0.03±0.01** |
| 40670 | **0.04±0.01** | 0.05±0.01 | **0.04±0.01** | 0.04±0.02 | **0.03±0.01** | **0.03±0.01** | **0.03±0.01** |
| 40685 | **0.0±0.0** | **0.0±0.0** | **0.0±0.0** | **0.0±0.0** | **0.0±0.0** | **0.0±0.0** | **0.0±0.0** |
| 40701 | 0.05±0.01 | **0.04±0.01** | 0.05±0.01 | 0.05±0.01 | **0.04±0.0** | 0.05±0.01 | 0.05±0.01 |
| 40900 | **0.01±0.0** | **0.01±0.0** | **0.01±0.0** | **0.01±0.0** | **0.01±0.0** | **0.01±0.0** | **0.01±0.0** |
| 40975 | **0.0±0.0** | **0.0±0.0** | **0.0±0.0** | **0.0±0.01** | **0.0±0.0** | 0.01±0.01 | 0.01±0.01 |
| 40978 | **0.02±0.01** | 0.03±0.01 | **0.02±0.01** | 0.03±0.01 | 0.03±0.01 | 0.03±0.01 | 0.03±0.01 |
| 40981 | 0.14±0.05 ○ | 0.14±0.03 ○ | **0.12±0.03** | 0.13±0.03 | 0.12±0.03 | 0.12±0.03 | 0.12±0.02 |
| 40982 | **0.19±0.02** | 0.2±0.02 | 0.2±0.03 | 0.21±0.03 | 0.21±0.04   † | 0.21±0.02 | 0.25±0.1 |
| 40983 | **0.02±0.0** | **0.02±0.0** | **0.02±0.0** | 0.02±0.01 | **0.01±0.0** | **0.01±0.0** | **0.01±0.0** |
| 40984 | **0.02±0.01** | 0.04±0.01 ○ ○ | **0.02±0.01** | 0.02±0.01 | **0.02±0.01** | **0.02±0.01** | **0.02±0.01** |
| 41027 | 0.16±0.02 ○ ○ | 0.15±0.02 ○ ○ | 0.09±0.0 | 0.09±0.02 | **0.07±0.01** † † | 0.16±0.01 | 0.14±0.0 |
| 41138 | **0.01±0.0** | **0.01±0.0** | **0.01±0.0** | **0.01±0.0** | **0.01±0.0** | **0.01±0.0** | **0.01±0.0** |
| 41142 | 0.29±0.02 ○ ○ | 0.28±0.02   ○ | 0.27±0.02 | 0.26±0.03 | **0.27±0.03** | 0.28±0.02 | 0.28±0.02 |
| 41143 | **0.16±0.0** | 0.17±0.02 | 0.17±0.02 | 0.18±0.02 | **0.18±0.02** | 0.19±0.02 | 0.2±0.04 |
| 41144 | 0.27±0.01 | 0.19±0.02   ○ | 0.21±0.05 | 0.09±0.01 | **0.1±0.02** † | 0.24±0.02 | 0.11±0.02 |
| 41145 | 0.29±0.03 ○ ○ | 0.26±0.02   ○ | 0.25±0.01 | **0.16±0.03** | **0.18±0.01** † † | 0.25±0.02 | 0.24±0.01 |
| 41146 | **0.04±0.0** | 0.07±0.01 ○ ○ | 0.05±0.01 | 0.04±0.01 | 0.05±0.01 | 0.05±0.01 | 0.04±0.01 |
| 41147 | nan | **0.32±0.01** | 0.33±0.01 | 0.32±0.0 | nan | **0.32±0.0** | nan |
| 41150 | **0.07±0.0** | 0.08±0.01 | 0.09±0.0 | 0.08±0.01 | **0.06±0.0** | 0.07±0.0 | **0.06±0.0** |
| 41156 | **0.15±0.03** | **0.15±0.02** | **0.15±0.01** | **0.15±0.01** | **0.14±0.01** | **0.14±0.01** | **0.14±0.01** |
| 41157 | nan | 0.21±0.14 ○ | **0.15±0.09** | 0.22±0.11 | 0.24±0.09 ○ | **0.11±0.09** | 0.26±0.15 |
| 41158 | **0.07±0.01**   ● | **0.07±0.01**   ● | **0.07±0.04** | 0.19±0.05 | **0.05±0.01** | **0.05±0.01** | **0.05±0.01** |
| 41159 | nan | 0.2±0.01 | 0.21±0.01 | **0.18±0.03** | **0.17±0.01** | 0.18±0.01 | nan |
| 41162 | **0.1±0.0** | **0.1±0.0**   ● | **0.1±0.0** | 0.22±0.06 | **0.1±0.0** | **0.1±0.0** | **0.1±0.0** |
| 41163 | 0.08±0.0 | 0.03±0.0 | 0.03±0.0 | **0.02±0.01** | **0.01±0.0** | 0.02±0.0 | 0.02±0.0 |
| 41164 | 0.35±0.01 ○ ○ | 0.32±0.02 ○ | **0.3±0.01** | 0.33±0.02 | 0.31±0.02 ○ ○ | **0.29±0.01** | **0.29±0.01** |
| 41165 | nan | 0.62±0.02 | 0.6±0.03 | nan | **0.55±0.02** | **0.55±0.01** | 0.56±0.01 |
| 41166 | 0.36±0.03 | 0.35±0.01 ○ | **0.3±0.01** | 0.34±0.1 | **0.28±0.0** † † | 0.33±0.0 | 0.32±0.0 |
| 41167 | nan | 0.96±0.0 ○ ○ | 0.12±0.07 | **0.09±0.0** | 0.31±0.0 | nan | nan |
| 41168 | 0.34±0.01 | 0.3±0.0 ● | 0.34±0.01 | **0.29±0.01** | **0.27±0.0** ● ● | 0.29±0.01 | 0.29±0.0 |
| 41169 | **0.71±0.0** | 0.72±0.0   ● | 0.72±0.0 | 0.75±0.01 | 0.69±0.01 ○ ○ | 0.65±0.0 | **0.64±0.0** |
| 42732 | nan | **0.12±0.0** | **0.12±0.0** | **0.12±0.0** | **0.12±0.0** | **0.12±0.0** | **0.12±0.0** |
| 42733 | **0.16±0.0** | 0.17±0.02   ● | 0.17±0.0 | 0.25±0.03 | **0.17±0.0** | **0.17±0.0** | **0.17±0.0** |

