# OpenReview forum: "Replacing the Ex-Def Baseline in AutoML by Naive AutoML"
_ICML.cc/2021/Workshop/AutoML — AutoML@ICML2021 Poster_

### Official Review · Reviewer_AXQ1 · 2021-06-14

**Rating:** 5
**Confidence:** 4

**Review:**

This paper introduces a simple baseline for assessing AutoML systems. The proposed approach, termed Naive AutoML, is based on an iterative tuning of predefined default ML pipeline. Each component of the default pipeline (probing, feature scaling, ..., tuning) is optimized separately following a fixed order.


The idea proposed by the paper is undoubtedly valuable for the AutoML  community. However, the paper has still some major issues (misleading claims, presentation of results). In its current form, I recommend a weak reject.


Pros
--------

The work is interesting as it highlights a huge issue on benchmarking AutoML systems. Indeed, Table 2 suggests that the improvements
of both Auto-Sklearn and Auto-Weka are not significant compared to the simple and easy to implement approach.
It may due to the fact that either some datasets are too easy or the time budget is not sufficient enough for Bayesian Optimization algorithms.
It thus important to explore different benchmarking setups (filtering simple datasets, extending time budget) to better show the merits of
advanced approach. IMO, the proposed Naive AutoML can then be used to quantify the quality of the benchmark.




Cons
-------

* "Surprisingly, the baselines used to assess the performance of AutoML tools are typically only other AutoML tools but no “simple” baselines."
This sentence is not true. Random search is widely used to assess AutoML systems and yet it is even more simplistic than Naive AutoML. It is not clear why the authors did not consider random search approach in their experiments. As far as I understand, Naive AutoML acts like  random search with the only difference that the components are optimized sequentially in Naive AutoML.
Adding random search baseline will strengthen the analysis and motivate the need of Naive AutoML.


* "one advantage of Naive AutoML over black-box optimization that already becomes clear here is that it directly generates important insights that can significantly support the data scientist working with it."
If one consider BO algorithm e.g. SMAC, one can easily generates the same insight as mentioned in the paper by investigating the underlying surrogate model. Even more, one can always provide analysis on the set of executed pipelines. I recommend the authors to either revise their claims or give example of analysis that would not be feasible for black-box optimization.


* It is hard to generalize the presented insights for other time budgets. The presentation of results can be improved by showing the comparison of different approaches over training time.
It is important to take into account the speed when all baselines reach the optimum.

---

### Official Review · Reviewer_P9Ao · 2021-06-17

**Rating:** 7
**Confidence:** 5

**Review:**

# Summary

The paper proposes a new simple baseline for AutoML which follows the workflow of many data scientist. It shows a strong performance on a short optimization budget (1h) and very short algorithm pipeline runs (1min) compared to Auto-Weka and Auto-Sklearn.

# Pros

* the paper is overall well written and easy to follow.
* the underlying idea is indeed simple and could establish another baseline for comparing AutoML tools, too.
* strong results against Auto-Weka and Auto-Sklearn

# Cons

* the approach is designed for ML pipeline problems and is not directly applicable to pure HPO or NAS tasks
* a comparison to random search is missing; however, I would expect that Auto-Weka and Auto-Sklearn perform at least as well as RS and therefore, I would not expect that the results change much.
* It is not clear why we need yet another baseline. Also I like the idea of Naive AutoML; I would not frame it as a baseline, but as a first step towards a conceptional simple but powerful AutoML approach
* the time limit for each algorithm run (1min) is very short

# Minor

* It's unclear to me whether $\theta$ are the parameters of the model or the hyperparameters of the algorithm
* \citep{} should be used to have parentheses around most refs
* some datasets in the experiments were removed because of some "data loading" issues.
* if the new version of auto-sklearn was used, the new auto-sklearn should be cited
* It's weird to have the main result table in the appendix. Adding at least a summarizing plot in the main paper would improve readability.
* It's unclear how the performance of Naive AutoML depends on the complexity of the search space. A more thorough study would improve the significance of the results
* The term "transformers" is very well established in the DL community. Although the term is well used here, it might raise other expectations. I would recommend to use a different term.
* "grid seach which is however not applicable in practice" Although I agree that grid search is very weak in most cases, I wonder why it is not applicable.

---

### Official Review · Reviewer_cLSt · 2021-06-18
**Well-written scathing rebuke of some SOTA AutoML systems, with some room for improvement**

**Rating:** 8
**Confidence:** 4

**Review:**

This is a well-written paper that offers a potentially scathing rebuke of a handful of the state-of-the-art AutoML systems if the findings hold up. Notably, the authors highlight that a simple analysis pipeline “optimization” approach (akin to writing a “for” loop over a handful of default transformers and predictors) matches and even sometimes outperforms a handful of state-of-the-art AutoML systems on a variety of established classification problems. The authors also go on to suggest that this Naive AutoML approach should be considered a standard baseline for comparison in the future for AutoML. I generally agree with the author’s premise that all AutoML papers should include a simple baseline for comparison, given how powerful the primitives are that we are optimizing over.

Although I find the premise of this paper interesting and thought-provoking, I do hold some qualms with the methodology and presentation in the paper, which I describe in detail below. All that said, I believe that this paper is well worth discussing at the AutoML workshop this year.

First and foremost, I find it disappointing that the authors chose to leave TPOT out of the comparison to their Naive AutoML system in this paper. I say this not because I’m the original TPOT author, but rather because leaving out TPOT (or an AutoML system similar to TPOT) leaves out an entire important class of AutoML systems that allow for flexible analysis pipeline structures and are based on evolutionary optimization techniques. The author could greatly strengthen this paper by including TPOT in the comparison, and may indeed strengthen their premise and findings by showing that these findings apply beyond the constrained pipeline search space.

Comparing to a random search would be valuable as well, given that prior work has shown random search to perform equally well or better in the hyperparameter optimization domain. e.g., https://www.jmlr.org/papers/volume13/bergstra12a/bergstra12a.

I found Table 2 in the Supplemental Information to be rather impenetrable. I had to stare at the dense table of numbers for quite a while to make much of the table, and even then I struggled to find a broader take-away at first glance. Table 2 should be replaced with a visualization of the results, e.g., box plots showing the median and distribution of performance of each system on each dataset.

Lastly, most of the performance differences among systems in Table 2 are quite small. In all cases where a “winner” is declared on a dataset, the difference is usually no greater than ~0.05 error (with few exceptions), which could easily be noise and may not even be statistically significant. For that reason, the interpretation of this paper may be that AutoML needs better baseline datasets, not better baseline models to compare AutoML systems to. Indeed, it may be the case that most of these datasets are practically “solved” because the supplied features are not rich enough to reach 0 error, and any progress made by AutoML systems on those datasets is simply finding better ways to overfit on that dataset. The author should consider including simulated datasets in their comparison (e.g., PMLB https://biodatamining.biomedcentral.com/articles/10.1186/s13040-017-0154-4), where the relationship between features and target is known and tunable.

---

### Decision · Program_Chairs · 2021-06-21

Accept (Poster)